# *Tuta absoluta*-Specific DNA in Domestic and Synanthropic Vertebrate Insectivore Feces

**DOI:** 10.3390/insects14080673

**Published:** 2023-07-28

**Authors:** Dirk Janssen, Emilio González-Miras, Estefanía Rodríguez

**Affiliations:** 1Department of Sustainable Crop Protection, Andalusian Institute of Agricultural and Fisheries Research and Training (IFAPA), Paraje San Nicolás, Autovía del Mediterráneo, Exit 420, E-04745 La Mojonera, Almeria, Spain; dirk.janssen@juntadeandalucia.es; 2SERBAL (Sociedad para Estudio y la Recuperación de la Biodiversidad Almeriense), E-04720 Almeria, Spain; emiliogmiras@serbal-almeria.org

**Keywords:** bats, birds, greenhouse, insectivores, lizards, qPCR, tomato, *Tuta absoluta*

## Abstract

**Simple Summary:**

The tomato leaf miner, *Tuta absoluta*, is one of the most harmful pests to greenhouse tomato crops in the Mediterranean. The biological control of this pest is based on parasitoid and predator insects. However, it may be worthwhile to measure whether the pest is part of the diet of domestic and synanthropic vertebrates like birds, bats, and lizards. We carried out our research in Southern Spain, an area well-known for its extensive tomato farming. TaqMan real-time PCR was used to find *T. absoluta* in domestic and synanthropic vertebrate feces. The efficiencies of three different DNA extraction methods were also compared. Our research demonstrates that in addition to domestic birds, bats, lizards, and insectivorous birds also consume *T. absoluta* and may offer an ecosystem service that merits further study.

**Abstract:**

The ecology of greenhouse pests generally involves parasitoid or predatory insects. However, we investigated whether the leaf miner *Tuta absoluta* (Meyrick, 1917) (Lepidoptera: Gelechiidae) is part of the diet of domestic and synanthropic vertebrate animals, such as birds, reptiles, and mammals, and that take part in an ecosystem that contains a high density of tomato greenhouses. Feces from domesticated partridges, common quails, and chickens, as well as from wild lizards were collected within tomato greenhouses, and fecal pellets from bats, swallows, common swifts, and house martins living in the vicinity of tomato greenhouses were collected outside. The efficiencies of three different DNA extraction methods were compared on bird, reptile, and mammal stool samples, and the DNA extracts were analyzed using probe real-time PCR for the presence of *T. absoluta* DNA. The results showed that bats fed on the pest, which was also part of the diet of several bird species: partridges and common quails kept within tomato greenhouses and swallows and common swifts living outside but in the vicinity of tomato greenhouses. In addition, fecal samples of three lizard species living near tomato crops also tested positive for *T. absoluta* DNA. The results suggest that aerial foraging bats and insectivorous birds are part of ecosystems that involve leaf miners and tomato greenhouses.

## 1. Introduction

Currently, the tomato (*Lycopersicum esculentum* L.) is one of the most cultivated vegetables in the world and the one with the highest economic value, exceeding 182 million tons in 2017. The leading tomato-producing countries globally are China, European Union, USA, and Turkey [1], and its demand, cultivation, economic value, and trade are increasing continuously. Consequently, investments increase in research and development to improve productivity, quality, and resistance to pests and diseases, and also because during the last decades no other intensive vegetable has suffered so many phytosanitary problems with the emergence of new diseases and pests. Some common tomato arthropod pests are whiteflies (Hemiptera: Aleyrodidae), thrips (Thysanoptera: Thripidae), spider mites (Acari, Tetranychidae), tomato russet mites (Acari: Eriophyidae), aphids (Hemiptera: Aphididae), flea beetles (Coleoptera: Chrysomelidae), cutworm (Lepidoptera: Noctuidae), tomato hornworm (Lepidoptera: Sphingidae), and leaf miners like *Lyriomyza* spp. (Diptera: Agromyzidae) and *Tuta absoluta* (Meyrick, 1917) (Lepidoptera: Gelechiidae) [2,3].

The adult of *T. absoluta* is a moth with a wingspan of around one centimeter. Its larva feeds on tomato plants, producing large galleries in leaves, burrowing in stalks, and consuming apical buds and green and ripe fruits. The pest is capable of causing a yield loss of 100% [4]. Tomato is the main host plant, but *T. absoluta* also attacks other solanaceous crops such as eggplant (*Solanum melongena* L.), pepper (*Capsicum annuum* L.), potato (*Solanum tuberosum* L.), and tobacco (*Nicotiana tabacum* L.). Under favorable climatological conditions, eight to ten generations can occur in a single year, during which the pest exhibits deuterotokous parthenogenesis resulting in a very high population [5]. This invasive pest species is native to South America that has undergone a massive expansion since 2006 reaching large portions of Central America, the Caribbean, Africa, and Eurasia [6]. Important factors that lead to such establishment and expansion were identified as high reproductive capacity [5,7], resistance to pesticides such as pyrethroids and diamide [8], transport through global and regional trade [9], broad host range [10], widespread production of tomatoes [11], and the absence of effective surveillance methods in some areas [12].

Due to the destructive effects *T. absoluta* has on tomato crops, pest control methods are critical. Its larvae spend most of their time within the leaf or in the fruit, and thus, chemical control is difficult and demands up to five sprays per week and 36 times per production cycle of 12 weeks to be effective. This results in the development of resistance to pesticides, environmental pollution, and human health risks [13]. Biological control of *T. absoluta* is considered to be one of the alternatives, but finding an effective natural enemy for a new, invasive pest is not an easy task as a pest often has tens to hundreds of species attacking it [14]. *T. absoluta* is associated with almost 200 species of predators and parasitoids in their native and newly invaded areas [6,15]. Currently, four species of all pest-associated natural enemies are commercially used: the egg parasitoids *Trichogramma pretiosum* Riley and *Trichogramma achaeae* Nagaraja & Nagarkatti (Hymenoptera: Trichogrammatidae), and the predatory mirids *Nesidiocoris tenuis* (Reuter) and *Macrolophus pygmaeus* (Rambur) (Hemiptera: Miridae) [6,15]. In the Spanish province of Almeria, natural enemies are used in 64% of tomato crops [16].

Thus, biological control of this greenhouse pest is generally associated with parasitoid or predatory insects. However, vertebrate animals, such as birds, provide important ecosystem services, including important pest control effects on productive systems [17]. The typically low bird diversity observed in intensive agricultural landscapes renders them more susceptible to pests that cause important economic losses [18]. Although these pests have traditionally been controlled using chemical methods, recent work suggests that bird-mediated biological control is an effective and environmentally friendly form of ecological intensification practice [18]. In addition, Kuhl’s pipistrelle bats (*Pipistrellus kuhlii)* were recently found to exploit pink bollworm irruptions in cotton fields by opportunistic feeding. It has been suggested that synanthropic bats provide important pest suppression services, may function as conservation biological control (CBC) agents of cotton pests, and potentially contribute to suppressing additional deleterious arthropods found in their diet in high frequencies. Interestingly, *T. absoluta* was also found to be part of the diet of Kuhl’s *pipistrelle* [19].

In the present paper, we investigate whether *T. absoluta* is part of the diet of domestic and synanthropic vertebrate animals, such as birds, reptiles, and mammals, and that take part in an ecosystem that contains a high density of tomato greenhouses. First, we compared the efficiency in isolating general and species-specific DNA of three different extraction methods on stool samples form birds, reptile, and mammal, spiked with homogenates from adult *T. absoluta* leaf miners. Second, stool samples from partridges artificially fed with *T. absoluta* adults were used to determine if the DNA from the leaf miner could be found in subsequently collected feces. And third, feces from domesticated partridges, common quails, and chickens, as well as from wild lizards, were collected within tomato greenhouses, and fecal pellets from bats, swallows, common swifts, and house martins living in the vicinity of tomato greenhouses were collected outside. DNA was extracted and analyzed using probe real-time PCR for the presence of *T. absoluta* DNA. The results confirm that bats feed on *T. absoluta* and show for the first time that this pest also is part of the diet of several bird and lizard species.

## 2. Materials and Methods

### 2.1. Insects and Feces Collection

The study was carried out in the province of Almería (South–East Spain), one of the biggest vegetable production regions in Europe (Figure 1) [20]. For insects and feces sampling, we considered two major regions of vegetable production in Almería, hereinafter denoted the East (A in Figure 1) and Southeast (B in Figure 1). The two areas present contrasting production conditions in terms of landscape context, crop type, vegetable production systems, and therefore, pest pressure. The East is an area characterized by vegetable crops cultivated in the open field, mainly lettuce (*Lactuca sativa* L.), broccoli (*Brassica oleracea* var. *italica*), and potato, mixed with citrus orchards (37°16′ N, 1°51′ W) (A in Figure 1). The Southeast is a highly and intensively cultivated greenhouse area, mainly with tomato, pepper, eggplant, cucumber (*Cucumis sativus* L.), zucchini (*Cucurbita pepo* L.), melon (*Cucumis melo* L.), and watermelon (*Citrullus lanatus* L.), and presents a high natural occurrence of *T. absoluta* populations (36°49′ N, 2°30′ W) (B in Figure 1).

All the samples were collected in the spring and early summer of 2022. Whole specimens of fifty *T. absoluta* adults were collected from black sticky traps (25 × 40 cm, Agrobio S.L. La Mojonera, Almería, Spain) combined with the *T. absoluta* Russel^®^ sex pheromone in several tomato greenhouses in the Southeast area. Specimens were carefully removed with forceps and maintained at −20 °C in 2 mL tubes until molecular analysis. To determine whether the pest under study was being consumed by vertebrates species inhabiting inside greenhouses, stool samples from lizards, partridges, chickens, and common quails were collected from organic tomato greenhouses in the Southeast area (B in Figure 1). To determine whether the pest was being consumed outside greenhouses, stool samples from nesting boxes and colonies of bats (*Pipistrellus pipistrellus*), nests of swallows (*Hirundo rustica*), house martins (*Delichon urbicum*), and common swifts (*Apus apus*) were also collected in the Southeast area (B in Figure 1). To determine whether the pest was being consumed in areas supporting less pest pressure, and therefore, far away from greenhouses, stool samples were also collected from nesting bird boxes and colonies of bats in the East (A in Figure 1). In all cases, the samples were stored at −20 °C until molecular analysis. For the DNA extraction method comparisons (see below), stools of central bearded dragon (*Pogona vitticeps*) were obtained from a local exotic pet shop and of healthy partridges from a local indoor farm. For the same purpose, feces from swallow and *P. pipistrellus* bats were collected in the wild.

### 2.2. DNA Extraction, Quantification and Quality Control

Tuta absoluta adults were homogenized individually using sterile pestles in 100 µL of 5% Chelex^®^ 100 Resin (Merck-Sigma-Aldrich, Madrid, Spain) [21]. The homogenates were briefly vortexed and incubated for 15 min at 54 °C followed by 3 min at 100 °C. Next, the homogenates were cooled for 5 min at −20 °C and then centrifuged for 5 min at 14,000× rpm.

The supernatant was removed and stored at −20 °C. For DNA extractions from vertebrate feces, DNA extraction kits were evaluated following the provided manuals: PowerFecal DNA Isolation Kit (Mo Bio Laboratories Inc., Carlsbad, CA, USA) and IndiSpin Pathogen Kit (Indical Bioscience, Leipzig, Germany). In addition, the Chelex method was also tested on the fecal samples. In a comparative study, feces from healthy partridge, swallow, *Pipistrellus* bat, and central bearded dragon were crushed and ground with a porcelain mortar and pestle. The dry powder was dissolved as 0.01% solutions in Milli-Q Water. Two hundred µL volumes of these samples then entered the extraction process and the eluates from the extractions were finally evaluated quantitatively estimating the DNA concentration using NanoDrop 2000 Ver. 1.6 spectrophotometer (Thermo Fisher Scientific, Waltham, MA, USA).

The evaluation was repeated with feces homogenates from the four sources, spiked with 15 µL of water containing 5 µg of extracted *T. absoluta* DNA, and the performance of the three methods was tested using real-time PCR. The extraction kit yielding the highest DNA concentration and *T. absoluta* DNA was further used for fecal DNA extraction from the domestic and synanthropic animal feces in this paper.

### 2.3. Detection of Tuta absoluta in Domestic and Wild Animals

Two partridges, held in captivity, were fed with grain food supplemented with *T. absoluta* adults. During the two days, the feces were collected and stored at −20 °C before use. These samples as well as feces collected at different points in the coastal area of the province of Almeria were homogenized and 0.3 g was mixed in 6 mL of water. The Indispin method was used to extract 200 µL volumes of the mixtures as described above. The amount of extracted DNA was estimated using the NanoDrop 2000 Ver. 1.6 spectrophotometer, and the presence of specific *T. absoluta* DNA was determined using probe real-time PCR.

### 2.4. Real-Time PCR and Sensitivity Analysis

Quantitative detection of *T. absoluta* DNA was conducted using a probe-based real-time PCR, adapted from the method described by Zink et al. [22]. One diagnostic primer pair, Ta-ITS-191F/Ta-ITS-345R, was used together with probe Ta-ITS-286P that was synthesized with a 5′ FAM fluorophore and dual quenching with a terminal 3′ BHQ1 quencher (Table 1). As an internal control, we utilized a previously designed 18S rDNA control probe ([23]; Table 1) using Cyanine (Cy-)5 fluorophore instead of Quasar 670, and a terminal 3′ BHQ3 quencher. We performed a duplex real-time PCR reaction as described by Ledezma et al. [24] to ensure the DNA extractions were of PCR quality. The real-time PCR reagent was qMAXSen™ Green Dye qPCR Master Mix2x (CANVAX, Cordoba, Spain), and determinations were conducted in a LightCycler 96 (Roche, Madrid, Spain) with the following protocol: (1) 95 °C for 30 s; (2) 95 °C for 10 s; (3) 60 °C for 30 s followed by data capture; and (4) repeat steps 2 and 3—40x. The reactions were carried out using 500 nM of all primers, 250 nM of probe, and 1 µL of DNA of varying concentration, in 96-well, thin-walled, white well hard-shell PCR plates (Thermo Fisher Scientific, Madrid, Spain) sealed with highly transparent sealing films (Brnd Gmbh, Wertheim, Germany).

A group of 10 *T. absoluta* individuals were ground in 400 µL Chelex solution (5%) and treated as described below. The assay sensitivity was tested by running serial dilutions of this *T. absoluta* DNA through the real-time PCR assay. DNA concentrations from 100 ng/µL to 0.0001 ng/µL were run using both the detection and control probes. Dilutions were prepared by adding appropriate amounts of water to a known starting concentration of DNA. The results of three independent runs were averaged and the quantitation cycle (Cq) values were compared to DNA concentration on a logarithmic scale to determine the slope, y-intercept, and correlation effects of DNA concentration on assay sensitivity [23].

## 3. Results

### 3.1. Real-Time PCR and Sensitivity Analysis

Concentrations of DNA from 100 ng/µL to 0.000001 ng/µL, isolated from adult *T. absoluta* individuals, were run using both the detection and control probes. The Cq values for the internal control ranged from 21.35 to 39.05 and for the *T. absoluta* DNA from 19.23 to 38.33. The average Cq values after three independent runs correlated with the logarithmic scale of the DNA concentrations down to 0.0001 ng/μL and 0.00001 ng/μL of DNA for the internal control (Cy5 probe) and the specific detection of *T. absoluta* DNA (FAM probe), respectively (Figure 2). The 18S control probe consistently produced a higher Cq value than the diagnostic FAM probe with the ΔCq (i.e., Cy5 − FAM Cq) ranging from −0.12 to 3.43 (mean 1.81 ± SD 1.07), indicating the increased sensitivity of the diagnostic probe over the control probe. When run in duplex, the standard curve for the two probes showed an increase in Cq values as DNA concentration decreased (Figure 2). There were no indications of possible interactive or negative effects of duplexing the probes. Based on the standard curve, samples with a DNA concentration ≥ 0.01 ng/µL should generate results suitable for analysis.

### 3.2. DNA Yield and Efficiency of Tuta absoluta DNA Purification

The three different methods that were tested for DNA extraction using feces from the bat, partridge, swallow, and central bearded dragon, each had unique features based on the inclusion of physical disruption of the samples (beads in PowerFecal, mortar and pestle in Chelex), chemical (proteinase K in Indispin), heat (100 °C in Chelex), and ethanol precipitation of DNA (PowerFecal and Indispin) (Table 2). In two separate tests, the PowerFecal method produced the lowest eluate DNA concentrations, varying from 2.5 to 17.9 ng/μL. The Indispin method yielded from 21.9 to 800.5 ng/μL DNA, and the Chelex method from 503.9 to 1683.2 ng/μL, depending on the source of feces; the Indispin method yielded the least amount of DNA from bat feces and the most amount from swallow feces, whereas the Chelex method yielded the lowest amount of DNA from central bearded dragon (*P. vitticeps*) and the highest amounts from swallow (*H. rustica*) (Table 3).

The PowerFecal method also produced the lowest eluate DNA concentrations, specifically of *T. absoluta*, that was spiked into the feces before extraction. Cq values of the real-time PCR varied between 23.8 and 29.46 (FAM dye probe). The Chelex method was slightly better (19.42 to 29.08 Cq) whereas the Indispin method yielded the best Cq values, ranging between 15.57 (for *P. perdix*) and 24.83 (for *H. rustica*) (Table 3). The Indispin Pathogen Kit was the method of choice for evaluating the collected stool samples from domestic and wild animals associated with tomato greenhouses.

### 3.3. Tuta absoluta DNA in Domestic and Wild Vertebrate Animals

*T. absoluta* DNA was readily detected in feces from two partridges, fed artificially with adults from this leaf miner; Cq values of the real-time PCR analyses using the FAM-labelled probe were 25.60 and 29.25, respectively, whereas feces from partridges not fed with *T. absoluta* (healthy controls) were negative (results not shown). This result showed us that DNA from the pest was detectable in feces from birds fed with the pest. Subsequently, feces DNA from domestic and wild animals in or close to tomato greenhouses was extracted using the Indispin Kit. Most showed positive Cq values albeit corresponding to different concentrations. The highest values were found in feces from bats and swallows (approx Cq 31), followed by feces from partridges, quails, swifts, and lizards (Cq 33-35). No *T. absoluta* DNA was detected in feces from house martins, chickens, two swallows, and one common swift stool sample (Table 4).

## 4. Discussion

We used Taqman real-time PCR to detect *T. absoluta* in the feces of vertebrates. Our test was adapted from Zink et al. [20] that amplifies DNA from this pest species, and not from other members of the family Gelechiidae such as the tomato pinworm *Keiferia lycopersicella.* The assay was adjusted for use in the Roche 96 Lightcycler, and with Cy5 dye for the 18S (control probe) instead of Quasar 670, and also different quenchers in the probes for the internal control and *T. absoluta* [22]. The resulting sensitivity of the assay was compatible and significant amplifications were obtained at concentrations down to 0.00001 ng/mL. This assay also readily detected *T. absoluta* DNA in DNA extractions of feces from birds, reptiles, and mammals. This detection, however, is highly dependent on efficient DNA extraction procedures that ideally would be suited for feces from these three groups of vertebrates. DNA extraction from avian feces is determined by the nature of birds mixing digestive residuals and urinary compounds to a single heterogeneous fecal deposit that may contain uric acid, bile salts, nucleases, and partly/non-degraded complex polysaccharides [25,26,27].

DNA Stool Mini Kit (Qiagen) has been recommended for the extraction of DNA from feces from bats [28], and from birds [29] and reptiles [30]. PowerFecal DNA Isolation Kit (MoBio) and derivatives from MoBio have been recommended for DNA extraction of feces from bats [19] and birds [31]. In addition, QIAamp Cador Pathogen Kit has been recommended for feces from birds [29]. Following the recommendations of Eriksson et al. [29], ‘feces-specific’ DNA extraction kits were discarded in the present study because they would perform very poorly, whereas QIAamp Cador Pathogen Kit eluates showed the best results. The QIAamp Cador Pathogen Kit is now continued as the Indispin Pathogen Kit (Indical Bioscience), which, for the present study, we compared with the PowerFecal DNA extraction (MoBio). In addition, we tested the Chelex method, which has been used specifically to extract *T. absoluta* DNA [32]. Our results showed that the MoBio DNA isolation kit produced the lowest eluate total DNA concentration. The Indispin method yielded more total DNA from bat, bird, and reptile feces, confirming the reports from Eriksson et al., [29]. The Chelex yielded the highest amounts of DNA method from 503.9 to 1683.2 ng/mL, depending on the source of feces; the Indispin method yielded the least amount of DNA from bat feces and the most amount from swallow feces, whereas the Chelex method yielded the lowest amount of DNA from *P. vitticeps* and the highest amounts from *H. rustica* (Table 3).

From an ecological point of view, feces from partridges, fed artificially with *T. absoluta,* contained DNA from this leaf miner that was detected using real-time PCR yielding Cq values of 25.60 and 29.25. These values were near to those found for feces collected from *P. pipistrellus* bats (e.g., 27.13) and swallows (e.g., 30.17), which suggests that the leaf miner is part of the diet of these wild mammals and birds. Lesser amounts of *T. absoluta* DNA were found in domesticated partridges that were reared inside a tomato greenhouse (Cq 33.92), and even fewer amounts were found in feces from the common swift (Cq 34.91) and from common quails that were reared inside a greenhouse (Cq 35.25). We detected *T. absoluta* DNA in feces from three lizard species found near tomato crops (Cq 34.44-35.47). Most lizards in arid habitats follow a wide foraging strategy. The Moorish gecko *Tarentola mauritanica*, a gecko frequently inhabiting humanized habitats, has been repeatedly classified as a sit-and-wait predator. Dietary analysis, however, suggests that the Moorish gecko is nocturnal and captures prey belonging to diverse taxonomic groups, mainly ground-dwelling arthropods, including lepidopteran larvae and adults [33]. Bats play a relevant action in the protection of economically important crops against lepidopteran pests [34]. This is the first report of *T. absoluta* predation by bats in this important tomato-producing area. Recently, Kuhl’s *pipistrelle* (*P. kuhlii*) was found to feed on *T. absoluta*. It has been suggested that synanthropic bats provide important pest suppression services, and may function as CBC agents of crop pests in the Eastern Mediterranean and potentially contribute to suppressing deleterious arthropods found in their diet in high frequencies [19].

Bats are opportunistic feeders and typically switch to whatever insects are available, moving over extensive areas and accumulating at outbreaks of pest insects [35,36]. Similarly, bat activity and the consumption of the pecan nut casebearer, *Acrobasis nuxvorella* (Lepidoptera: Pyralidae; Neunzig), coincided with peak outbreaks in orchards [37]. The consumption of the four main pest species by insectivorous bats in Macadamia was also correlated with pest outbreaks [38]. Fecal samples from six common bat species in Madagascar rice fields showed that bats foraged on economically important pests [39]. A continent-wide landscape analysis across southern Europe and DNA metabarcoding showed that the dietary patterns of the common bent-wing bat (*Miniopterus schreibersii*) are predicted by the agricultural intensification in the landscape [40]. In vineyards, lesser horseshoe bats (*Rhinolophus hipposideros*) eat a variety of pest species, and their diet changes seasonally [41]. Rodríguez-San Pedro et al. [42] have provided evidence that bats reduce grapevine pest insect infections and thus increase vineyard yield and winegrowers’ income. They conclude that bats should be included in future biodiversity conservation plans in vineyards and be considered within agricultural management strategies based on natural pest suppression. So, our results confirm the findings of Cohen et al. [19] that *Pipistrellus* sp. feeds on *T. absoluta,* but this time in the surroundings of tomato greenhouses in the Western Mediterranean.

Similar amounts of *T. absoluta* DNA were found in the feces of swallows, and this is the first report of this pest being found as part of the diet of synanthropic birds. *T. absoluta* DNA was also detected, but in fewer amounts in feces from common swifts, and none in those from house martins (Table 4). Common swifts, swallows, and house martins occur widely in the Western Mediterranean, breed sympatrically and most of their diet is obtained from agricultural sources of invertebrate prey [43]. It was quite surprising that bats and swallows’ predation on *T. absoluta* was so high and persistent over the fields, considering the setting in an intensively farmed semi-urban area. In comparison with bats, the predation by birds was variable (Table 4). The results could be partly explained by foraging niche differences in the use of different air layers—that is, foraging at different heights [44]. *T. absoluta* can move several kilometers by flying or drifting with the wind [45], but it is typically a low-flying insect found at lower layers near 50–60 cm above the ground [46,47,48], where swallows could feed on it [49]. This height is similar to that of the foraging niche of the bat *P. pipistrellus* [50]. Nevertheless, house martins have a mean fly height of 21 m [48] (Waugh, 1978) while swifts cover large distances to foraging grounds [51]. This could explain why *T. absoluta* constitutes a part of the diet of swallows and bats because predators and preys have overlapping flight heights. The relationship between flight height and foraging efficiency of crop pests of swallows, house martins, and swifts, and the potential benefits of all three bird species in terms of pest reduction, reduced crop damage, and improved crop yield, has also been proposed in field crops [52]. Moreover, the foraging activity of some bat species is significantly higher near pheromone lures, i.e., in areas of expected increased prey availability [53]. Therefore, the detection of considerable amounts of *T. absoluta* DNA in the feces of bats near tomato greenhouses is very significant. Sex pheromone-based biotechnical control of *T. absoluta* is an increasingly successful management practice in Almerian greenhouses, which could have influenced the foraging activity of bats species in our study area. The activity pattern of *T. absoluta* is usually nocturnal with adults usually remaining hidden during the day, showing greater morning–crepuscular activity with adults dispersing among crops by flying [4]. Our results indicate that crepuscular and nocturnal pests like *T. absoluta* are exposed to bat predation, not only by spatial co-occurrence (overlapping foraging niches) but also by merely temporal co-occurrence.

In contrast to domestic birds, this study shows that wild aerial foraging bats and insectivorous birds feed on *T. absoluta* and perhaps provide an ecosystem service worth being investigated. Our results suggest that in the control of the leaf miner in areas of tomato greenhouses, the potential integration of such bats and birds into pest management schemes might provide economic benefits. However, further studies are needed on the movements of *T. absoluta* outside and between greenhouses, and on the potential role of foraging vertebrates as agents of the biological control of these crop insect pests.

## 5. Conclusions

The leaf miner *Tuta absoluta* is a major pest of tomato. DNA from this insect was detected in feces from wild lizards and from domesticated partridges and common quails, that were collected within tomato greenhouses and also in fecal pellets from bats, swallows, and common swifts, living in the vicinity of tomato greenhouses that were collected outside. The results suggest that aerial foraging bats and some insectivorous birds species are part of ecosystems that involve leaf miners and tomato greenhouses.

## Figures and Tables

**Figure 1 insects-14-00673-f001:**
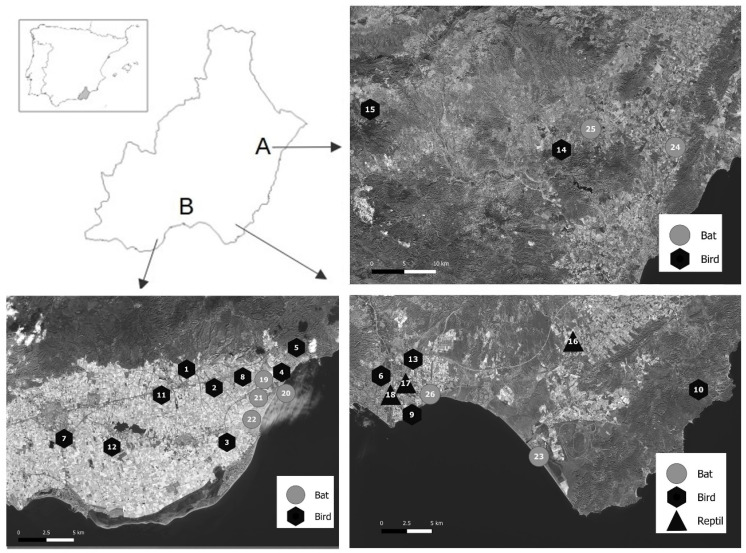
Location of samples of feces from birds, reptiles, and bats. Upper left: location of the province of Almeria in South–East Spain and two areas within (A: East and B: Southeast). Numbers refer to collected samples.

**Figure 2 insects-14-00673-f002:**
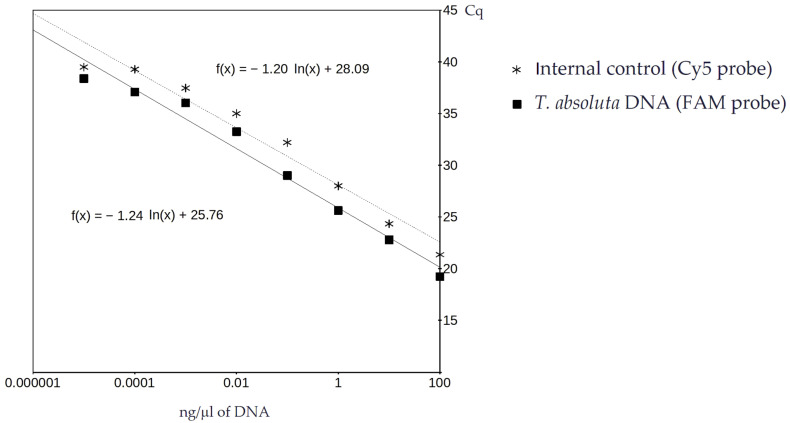
Standard curve of Cq values for serial dilutions of *T. absoluta* DNA, extracted from adult individuals, run with the real-time PCR assay.

**Table 1 insects-14-00673-t001:** Primers and probes used in this study.

Name	Description	Sequence	Tm (°C)	Source
Ta-ITS-191F	ITS (diagnostic forward primer)	5′-GCGAGCCAAGTGATCCAC-3‘	64.7	[20]
Ta-ITS-345R	ITS (diagnostic reverse primer)	5′-AGGCGACGATGTGTACGA-3‘	63.1	[20]
Ta-ITS-286P	ITS (diagnostic probe)	5′-FAM-CTTAGAATACAT-CGTGCAGGCAGC-BHQ-1-3‘	67.6	[20] ^1^
RT-18S-F2	18S (control forward primer)	5′-ACCGCCCTAGTTCTAACCGTAAA-3‘	65.4	[21]
RT-18S-R2	18S (control reverse primer)	5′-CCGCCGAGCCATTGTAGTAA-3‘	66.9	[21]
RT-18S-P2	18S (control probe)	5′-Cy5-TGTCATCTAGC-GATCCGCCGA-BHQ-3-3‘	72.1	[21] ^1^

^1^ The dye/quencher combinations are from the present work.

**Table 2 insects-14-00673-t002:** Critical features of the DNA extractions methods used on collected feces.

Treatment	DNA Extraction Methods		
	PowerFecal DNA Isolation Kit	Indispin Pathogen Kit	Chelex
Physical disruption	yes	no	yes
Proteinase K	no	yes	no
Boil (100 °C)	no	no	yes
Ethanol precipitation	yes	yes	no

**Table 3 insects-14-00673-t003:** Purification of general DNA and specific *T. absoluta* DNA from stools of Pipistrelle bat, partridge, swallow, and Iberian wall lizard.

Method	Origin	DNA ng/μL (mean ± SD) ^1^		Cq ^2^	
		Experiment 1	Experiment 2	FAM	Cy5
PowerFecal	*P. pipistrellus*	12.4 ± 0.7	17.9 ± 0.9	29.46	28.79
	*P. perdix*	5.4 ± 0.4	17.6 ± 2.3	25.99	27.05
	*H. rustica*	2.5 ± 1.5	13.9 ± 4.6	28.37	29.33
	*P. vitticeps*	4.0 ± 3.7	14.1 ± 3.7	23.8	21.95
Indispin	*P. pipistrellus*	21.9 ± 3.0	22.3 ± 2.1	21.47	22.17
	*P. perdix*	60.5 ± 2.6	101.3 ± 1.6	15.57	15.98
	*H. rustica*	309.5 ± 22.1	800.5 ± 14.3	24.88	25.34
	*P. vitticeps*	61.0 ± 3.9	53.6 ± 18.0	20.26	18.56
Chelex	*P. pipistrellus*	1160.4 ± 22.4	1358.7 ± 16.3	20.05	21.54
	*P. perdix*	1404.4 ± 42.8	1471.6 ± 6.9	28.01	27.79
	*H. rustica*	1269.9 ± 43.8	1683.2 ± 46.3	29.08	27.79
	*P. vitticeps*	503.9 ± 16.2	395.1 ± 7.2	19.42	23.02

^1^ Purification of DNA from stools of two birds (partridge *P. perdix* and swallow *H. rustica*), one mammal (*P. pipistrellus* bats), and one reptile (central bearded dragon *P. viticeps*) in two independent experiments. ^2^ The Cq values represent the specific detection of *T. absoluta* (FAM) and the internal control (Cy5) in real-time PCR from the second experiment in which samples were spiked with *T. absoluta* DNA.

**Table 4 insects-14-00673-t004:** Sources of collected feces, amounts of total DNA extracted, and specific *T. absoluta* DNA detected.

Class	Species		Location	Sample ID *	Total General DNA (*μ*g/uL)	Specific *T. absoluta* DNA (Cq)
	Latin Name	Common Name	Inside/Outside Tomato Greenhouse			
Aves	*Perdix perdix*	Partidge	Inside	1	91.0	33.92
	*Gallus domesticus*	Chicken	Inside	2	26.4	- **
	*Coturnix coturnix*	Common quail	Inside	3	126.9	35.25
	*Apus apus*	Common swift	Outside	4	508.6	34.91
	*id.*	*id.*	*id.*	5	832.0	34.70
	*id.*	*id.*	*id.*	6	450.4	-
	*Delichon urbicum*	House martin	Outside	7	774.1	-
	*id.*	*id.*	*id.*	8	726.9	-
	*id.*	*id.*	*id.*	9	642.9	-
	*id.*	*id.*	*id.*	10	501.3	-
	*id.*	*id.*	*id.*	11	747.7	-
	*Hirundo rustica*	Swallow	Outside	12	678.9	31.83
	*id.*	*id.*	*id.*	13	612.0	30.17
	*id.*	*id.*	*id.*	14	187.3	-
	*id.*	*id.*	*id.*	15	559.0	-
Reptilia	*Podarcis hispanicus*	Iberian wall lizard	Inside	16	66.8	35.47
	*Tarentola mauritanica*	Common wall gecko	Inside	17	187.9	34.44
	*Psammodromus algirus*	Algerian sand racer	Inside	18	411.3	35.03
Mammalia	*Pipistrellus pipistrellus*	Pipistrelle microbat	Outside	19	17.4	31.56
	*id.*	*id.*	*id.*	20	42.1	31.94
	*id.*	*id.*	*id.*	21	54.1	34.31
	*id.*	*id.*	*id.*	22	60.0	33.65
	*id.*	*id.*	*id.*	23	53.7	33.58
	*id.*	*id.*	*id.*	24	178.4	29.53
	*id.*	*id.*	*id.*	25	149.9	27.13
	*id.*	*id.*	*id.*	26	45.9	31.77

* Numbers refer to the locations in Figure 1; ** not detected; *id*. means *idem*

## Data Availability

Data is contained within the article.

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
