# Peer review of "Tuta absoluta-Specific DNA in Domestic and Synanthropic Vertebrate Insectivore Feces"

_insects, 2023, doi:10.3390/insects14080673_

Round 1
Reviewer 1 Report
As understanding biological control of agricultural pests is intensifying is extending beyond predators and parasitoids to vertebrate contribution to such control due to issues with chemical control, including the off target effects and resistance development. Interest in understanding the potential contribution of insectivores including bats, birds, reptiles is of increasing importance. However to be most valuable it is important to investigate not just the generalities but also in a given situation or area, to identify the relevant taxa as is done here This study makes an important contribution to this literature-yes bats, birds and reptiles have been shown to consume Tuta absoluta and the method section (and relevant comments in the discussion) should be very useful to other investigators.
There are a couple of places in the introduction where I would like to see more information-
L22 needs rewriting for clarity “in contrast to domestic birds ….
Meaning bats and insectivorous birds to consume …..?
L78-79 clarify that the natural enemies listed are indeed released in greenhouses (or in the field) and give some indication of the extent of their use and the success observed.
L83 this is a wide-ranging statement. if it refers to more than tomatoes in greenhouses this needs to be substantiated with expanded information and more references.
LL90 ff is it the case that the microbats in the area under investigation are in fact all insectivores? If so add this info then say for eg the bat specifically mentioned
L 105 italics for T. absoluta
L 502 is this reference 52?
Author Response
Dear reviewer, thank you very much your interesting comments and suggestions. We detail the changes made corresponding to your comments:
L22 needs rewriting for clarity “in contrast to domestic birds …. Meaning bats and insectivorous birds to consume …..?
Response -> This was our mistake. It should have been "in addition to" instead of "in contrast to". This has now been corrected.
L78-79 clarify that the natural enemies listed are indeed released in greenhouses (or in the field) and give some indication of the extent of their use and the success observed.
Response -> Thank you for the suggestions. We have added the degree of application (64% of tomato crops) in the text together with a recent bibliographic reference.
L83 this is a wide-ranging statement. if it refers to more than tomatoes in greenhouses this needs to be substantiated with expanded information and more references.
Response-> We agree. The sentence was too wide. We have corrected this a restricted it to Tuta absoluta and merely summarizes the previous paragraph in order to start a new argument.
LL90 ff is it the case that the microbats in the area under investigation are in fact all insectivores? If so add this info then say for eg the bat specifically mentioned
Response-> The bats discussed in the section LL90 ff are insectivores and further information is within reference 18 cited on line 97.
L 105 italics for T. absoluta
Response -> Done
L 502 is this reference 52?
Response-> Yes, it is. Thank you.
Reviewer 2 Report
Leaf miner DNA in feces
This manuscript presents some interesting material, but it needs a major rewrite. Some matters are trivial and easily addressed, some will require more complete modification.
Trivial matters:
1. The simple summary present results for birds and bats, but not what was found for lizards. This is not consistent with key results presented in the paper.
2. The use of term “ecosystem service” here seems unclear.
3. It is unclear why parts of the introduction (lines 83-87) discuss biological control and low bird diversity here in relation to the ability to detect the leaf minor DNA in feces.
4. Table 2 includes “poteinase K” as a reagent (proteinase K?).
More serious matters:
1. Figure 2 shows a standard calibration curve for Cq values, but there is no description that I can find about the nature of the leaf miner DNA used here.
2. Lines 181-185 talk about evaluation with feces that were “spiked” with leaf miner DNA without a clear explanation of why this was necessary or how it relates to experiments presented later in the paper where wild collected feces are analyzed.
3. The summary of results presented in lines 231-235 is confusing and difficult to follow. Is this all from the results using “spiked” DNA? How do these organisms relate to those reported on later?
4. A key set of results (Table 4) is mentioned in the results section, but the table itself does appear until the discussion section.
5. Much of the discussion section (lines 281-288, 289 – 297) read more like results.
6. Line 298 abruptly transitions from results to discussion like material, so perhaps this should begin a new paragraph.
Overall, there is good material here, but it needs work to be organized and presented more effectively.
Author Response
Dear reviewer, thank you very much your interesting comments and suggestions. We detail the changes made corresponding to your comments:
- The simple summary present results for birds and bats, but not what was found for lizards. This is not consistent with key results presented in the paper.
Response 1-> The results for lizards have now been included in L 22 in the simple summary
- The use of term “ecosystem service” here seems unclear.
Response 2-> Ecosystem services or eco-services are defined as the goods and services provided by ecosystems to humans (https://alus.ca/wp-content/uploads/2016/08/estimation-of-ecosystem.pdf). Per the 2006 Millennium Ecosystem Assessment (MA) (https://en.wikipedia.org/wiki/Millennium_Ecosystem_Assessment), ecosystem services are "the benefits people obtain from ecosystems". The MA also delineated the four categories of ecosystem services—supporting, provisioning, regulating, and cultural—discussed below. In simple terms pest control, provision of water, timber, fibers, and of medications. The argument here is that bats, lizards, and insectivorous birds do feed on pests such as T. absoluta on our foods, and that they possibly regulate the pest.
- It is unclear why parts of the introduction (lines 83-87) discuss biological control and low bird diversity here in relation to the ability to detect the leaf minor DNA in feces.
Response 3 -> This paragraph provides literature that supports the hypothesis that birds and bats could play a role in the biological control of crop pests. The first step in order to apply this hypothesis on tomato greenhouses and leaf miners, would be to investigate whether this pests is part of the diet of other animals besides the common predators and parasitoids. And this is investigated detecting the DNA of the prey in feces and fecal pellets.
- Table 2 includes “poteinase K” as a reagent (proteinase K?).
Response 4-> Thank you. this was a lapsus. It is proteinase K, indeed.
More serious matters:
- Figure 2 shows a standard calibration curve for Cq values, but there is no description that I can find about the nature of the leaf miner DNA used here.
Response 1-> We now have detailed the origin of the DNA in the title of the figure. The procedure to isolate the DNA is detailed in the materials and methods (LL168 ff) and the origin has now been detailed again also in the results section (LL 199 ff) where Figure 2 is cited.
- Lines 181-185 talk about evaluation with feces that were “spiked” with leaf miner DNA without a clear explanation of why this was necessary or how it relates to experiments presented later in the paper where wild collected feces are analyzed.
Response 2-> Spiking in analytical chemistry refers to the act of adding a known amount of a known substance to a sample in order to evaluate the performance of an analytical method. Or when citing a Thermo Scientific document "Matrix Spiking – Why Spike and How to Do It " https://static.fishersci.com/cmsassets/downloads/segment/Scientific/pdf/WaterAnalysis/Log112tipMatrixSpikeWhySpikeHowtoDoIt.pdf: (...) a matrix spike test helps answer the question “Are we getting good (valid) results when we use this method to test this sample or this type of sample?” A “good” matrix spike result increases our confidence in the accuracy and validity of the sample test results.
And applied on DNA extraction and detection: https://digitalcommons.fiu.edu/cgi/viewcontent.cgi?article=3020&context=etd Since we apply the extraction methods and qPCR detections on T. absoluta DNA in feces and fecal pellets from different animals, we wanted to make sure that these different matrices did not affect significantly the detection of the DNA.
- The summary of results presented in lines 231-235 is confusing and difficult to follow. Is this all from the results using “spiked” DNA? How do these organisms relate to those reported on later?
Response 3-> Yes, they are. The origin of the feces is mentioned in the section Materials and methods
L 138: "For the DNA extraction method comparisons (see below), stools of central bearded dragon (Pogona vitticeps) were obtained from a local exotic pet shop, and of healthy partridges from a local indoor farm." To complete this information we have now added the sentence: "For the same purpose, feces from swallow and P. pipistrellus bats, were collected in the wild." The legend of Figure 3 details that the feces represent two birds, one mammal (bat) and 1 reptile. All stools were spiked but the analyses yield general total DNA, and specific Tuta DNA as said in the title of Table 3.
- A key set of results (Table 4) is mentioned in the results section, but the table itself does appear until the discussion section.
Response 4-> We agree and we have now moved the Table 4 in the results sections where it is cited.
- Much of the discussion section (lines 281-288, 289 – 297) read more like results.
Response 5 -> In this section we briefly recall the results and contrast these with findings in the literature (references 27-30)
- Line 298 abruptly transitions from results to discussion like material, so perhaps this should begin a new paragraph.
Response 6-> The entire paragraph recalls quantitative (qPCR) results of Tuta DNA found in all animals, compares them (first half), and then disusses them with relevant references from the literature (second half).
Overall, there is good material here, but it needs work to be organized and presented more effectively.
..............................................................
Reviewer 3 Report
Dear Authors,
You have raised an important topic for research.
I have a few comments about your work.
In the title of the article, you can remove (Lepidoptera: Gelechiidae).
Introduction:
- the names of scientists in the Latin names of species are not written in brackets. In lines 42, 78-82. In lines 78-82, you have the names of the Order and Family in brackets, and the names of scientists are indicated.
- in lines 74-77. Check the sentence. There is a dot in the middle of a sentence, and a new sentence begins with a lowercase letter.
- lines 84-85. Maybe you should remove "in many ecosystems" (tautology).
Materials and methods:
- Correct section title to "2.1 Insects and feces collection"
- Sections 2.3 and 2.4 are best combined and placed before section 2.2. You first collect samples, then isolate DNA, and only then do PCR.
Results:
- Do not abbreviate the species name in the headings of Sections 3.2 and 3.3.
- In the results, the information is also presented in the wrong order. Your main task in the study is to identify the presence of moth DNA in the feces of various types of potential predators. However, only one paragraph is devoted to this issue in the results. It may be worth at least moving Table 4 to this section.
- Figure 1 in this section is completely out of place. You should put it in section 2.1 of Materials and Methods. In addition, it is necessary to increase the numbers in the figure, even when enlarged, they are not visible (Figure from the original image).
- What does [M3] mean in the title of figure 2. You need to reduce the size of the Figure or position it so that it is fully visible.
Discussion:
- what does "id." (or "id") mean in table 4? Give the explanation in the description.
- A lot of information is in the wrong place. Lines 257-270 are better presented in materials and methods. Lines 271-309 - in section 3.3 of the Results.
- in line 316 there is a typo in the words Madagascar
Reference:
Carefully check each source, they must be formatted in accordance with the requirements of the journal. Number 52 is completely lost. Doi are indicated differently.
Author Response
Dear reviewer, thank you very much your interesting comments and suggestions. We detail the changes made corresponding to your comments:
I have a few comments about your work.
In the title of the article, you can remove (Lepidoptera: Gelechiidae).
Response -> Thank you for the suggestion. We have now removed this from the title.
Introduction:
- the names of scientists in the Latin names of species are not written in brackets. In lines 42, 78-82. In lines 78-82, you have the names of the Order and Family in brackets, and the names of scientists are indicated.
Response -> Thank you for the suggestion. We have checked the taxonomic scientific names in the following link: https://fauna-eu.org/. A name following in brackets means that the name has been amended subsequent to first descriptions. We have now corrected this.
- in lines 74-77. Check the sentence. There is a dot in the middle of a sentence, and a new sentence begins with a lowercase letter.
Response-> Indeed, there is. We have now corrected this.
- lines 84-85. Maybe you should remove "in many ecosystems" (tautology).
Response-> yes, we have now removed that.
Materials and methods:
- Correct section title to "2.1 Insects and feces collection"
Response -> We have corrected this following the suggestion.
- Sections 2.3 and 2.4 are best combined and placed before section 2.2. You first collect samples, then isolate DNA, and only then do PCR.
Response-> We have moved section 2.3 and 2.4 before section 2.2 as suggested
Results:
- Do not abbreviate the species name in the headings of Sections 3.2 and 3.3.
Response -> We have corrected this now.
- In the results, the information is also presented in the wrong order. Your main task in the study is to identify the presence of moth DNA in the feces of various types of potential predators. However, only one paragraph is devoted to this issue in the results. It may be worth at least moving Table 4 to this section.
Response-> We agree and we have moved Table 4 to this section
- Figure 1 in this section is completely out of place. You should put it in section 2.1 of Materials and Methods. In addition, it is necessary to increase the numbers in the figure, even when enlarged, they are not visible (Figure from the original image).
Response-> We agree. We have moved Figure 1 near to section 2.1. Also, we have enlarged the numbers in the figure so they should now be easily visible.
- What does [M3] mean in the title of figure 2. You need to reduce the size of the Figure or position it so that it is fully visible.
Response -> We hace checked this issue, we think its refres to an editorial´ comment
Discussion:
- what does "id." (or "id") mean in table 4? Give the explanation in the description.
Response-> Idem is a Latin term meaning "the same". It is commonly abbreviated as id., It should be been put in italics, and this has now been corrected.
- A lot of information is in the wrong place. Lines 257-270 are better presented in materials and methods. Lines 271-309 - in section 3.3 of the Results.
Response-> Thank you for the suggestion. The information in lines 257-270 discusses the different extraction techniques and the relevant literature, applied on the fecal samples of the present paper. Also, the information in lines 271-309 recalls results but discussed with the relevant literature.
- in line 316 there is a typo in the words Madagascar
Response-> Thank you. this has been corrected
Reference:
Carefully check each source, they must be formatted in accordance with the requirements of the journal. Number 52 is completely lost. Doi are indicated differently.
Response-> We have corrected and formatted the references in accordance with the requirements of the journal. And the "lost reference" has no been integrated into the numbered list. Also, we have arranged the DOI typing.
Round 2
Reviewer 2 Report
This revised version of the manuscript is much improved. My only remaining suggestion would be to provide some additional clarification regarding the points I will make below.
It seems to me that overall, you have done 3 things that are of interest and value to the research community. These are:
1. You have evaluated 3 different DNA extractions methods for recovering DNA from fecal samples.
2. You have shown using RT-PCR that you can you can detect DNA from the leaf miner after either "spiking" it into the fecal samples or by adding it to material fed to birds.
3. You have shown that using the RT-PCR method, you can detect leaf miner DNA in wild collected fecal material from some insectivores, clearly indicating that it is part of their diet.
These are all valuable results, but I would still like to see some additional clarification in terms of explaining how your experimental design was used to evaluate these things in a stepwise manner.
Author Response
Dear reviewer, thank you very much for your comments. We totally agree that the study should be explained better. Following your suggestion we have now introduced new sentences explaining the rationale of research, presenting the stepwise experiments that were done. (By the way, it was PCR (detecting DNA), not RT-PCR (using to detect RNA). These changes were introduced in the last paragraph of the introduction section (Lines 101-106).
Reviewer 3 Report
Dear Authors,
The article got better after corrections. I have a few comments about your work.
line 174 - it would be better to give the name of the device as in the line 161,
lines 263-264 - numbers less than 10 should be written in words,
lines 360-367 - wrong formatting
Author Response
Dear reviewer, thank you very much for your useful comments.
We have introduced the following changes as to your specific comments:
line 174 - it would be better to give the name of the device as in the line 161,
Response: ok, done
lines 263-264 - numbers less than 10 should be written in words,
response: ok, done
lines 360-367 - wrong formatting,
response: ok, done. see lines 370-380